# DISTILLING KNOWLEDGE FROM READER TO RETRIEVER FOR QUESTION ANSWERING

**Gautier Izacard[1,2,3], Edouard Grave[1]**

[1]Facebook AI Research, [2]École normale supérieure, PSL University, [3]Inria
`{gizacard|egrave}@fb.com`

## ABSTRACT

The task of information retrieval is an important component of many natural language processing systems, such as open domain question answering. While traditional methods were based on hand-crafted features, continuous representations based on neural networks recently obtained competitive results. A challenge of using such methods is to obtain supervised data to train the retriever model, corresponding to pairs of query and support documents. In this paper, we propose a technique to learn retriever models for downstream tasks, inspired by knowledge distillation, and which does not require annotated pairs of query and documents. Our approach leverages attention scores of a reader model, used to solve the task based on retrieved documents, to obtain synthetic labels for the retriever. We evaluate our method on question answering, obtaining state-of-the-art results.

## 1 INTRODUCTION

Information retrieval is an important component for many natural language processing tasks, such as question answering (Voorhees et al., 1999) or fact checking (Thorne et al., 2018). For example, many real world question answering systems start by retrieving a set of support documents from a large source of knowledge such as Wikipedia. Then, a finer-grained model processes these documents to extract the answer. Traditionally, information retrieval systems were based on hand-crafted sparse representations of text documents, such as TF-IDF or BM25 (Jones, 1972; Robertson et al., 1995). Recently, methods based on dense vectors and machine learning have shown promising results (Karpukhin et al., 2020; Khattab et al., 2020). Deep neural networks based on pre-training, such as BERT (Devlin et al., 2019), have been used to encode documents into fixed-size representations. These representations are then queried using approximate nearest neighbors (Johnson et al., 2019). These techniques have lead to improved performance on various question answering tasks.

A challenge of applying machine learning to information retrieval is to obtain training data for the retriever. To train such models, one needs pairs of queries and the corresponding list of documents that contains the information corresponding to the queries. Unfortunately, hand-labeling data to that end is time consuming, and many datasets and applications lack such annotations. An alternative approach is to resort to heuristics, or weakly supervised learning, for example by considering that all documents containing the answer are positive examples. However, these approaches suffer from the following limitations. First, frequent answers or entities might lead to false positive examples. As an example, consider the question *"where was Ada Lovelace born?"*. The sentence *"Ada Lovelace died in 1852 in London"* would be considered as a positive example, because it contains the answer *"London"*. A second limitation is that for some tasks, such as fact checking or long form question answering, such heuristics might not be applicable directly.

In this paper, we propose a procedure to learn retriever systems without strong supervision in the form of pairs of queries and documents. Following previous work (Chen et al., 2017), our approach uses two models: the first one retrieves documents from a large source of knowledge (*the retriever*), the second one processes the support documents to solve the task (*the reader*). Our method is inspired by *knowledge distillation* (Hinton et al., 2015), and uses the reader model to obtain synthetic labels to train the retriever model. More precisely, we use a sequence-to-sequence model as the reader, and use the attention activations over the input documents as synthetic labels to train the retriever. Said otherwise, we assume that attention activations are a good proxy for the relevance of

documents. We then train the retriever to reproduce the ranking of documents corresponding to that metric.

We make the following contributions:

- First, we show that attention scores from a sequence-to-sequence reader model are a good measure of document relevance (Sec. 3.2) ;
- Second, inspired by knowledge distillation, we propose to iteratively train the retriever from these activations, and compare different loss functions (Sec. 3.4) ;
- Finally, we evaluate our method on three question-answering benchmarks, obtaining state-of-the-art results (Sec. 4).

Our code is available at: `github.com/facebookresearch/FiD`.

## 2 RELATED WORK

We briefly review information retrieval based on machine learning. We refer the reader to Manning et al. (2008) and Mitra et al. (2018) for a more exhaustive introduction to the subject.

**Vector space models.**    In traditional information retrieval systems, documents and queries are represented as sparse vectors, each dimension corresponding to a different term. Different schemes have been considered to weigh the different term, the most well known being based on inverse document frequency, or *term specificity* (Jones, 1972). This technique was later extended, leading to the BM25 weighting scheme which is still widely used today (Robertson et al., 1995). A limitation of sparse representations is that the terms of the query need to match the terms of the returned documents. To overcome this, Deerwester et al. (1990) proposed to use latent semantic analysis for indexing, leading to low-dimension dense representations of documents.

**Neural information retrieval.**    Following the success of deep learning for other natural processing tasks, neural networks were applied to the task of information retrieval. Huang et al. (2013) proposed a deep bag-of-words model, where queries and documents were embedded independently, a technique known as *bi-encoder*. Documents were then ranked by using the cosine similarity with the query, and the model was trained on clickthrough data from a search engine. This technique was later extended by using convolutional neural networks (Shen et al., 2014) and recurrent neural networks (Palangi et al., 2016). A limitation of independently embedding documents and query is that it does not capture fine-grained interactions between the query and documents. This lead Nogueira & Cho (2019) and Yang et al. (2019) to use a BERT model to jointly embed documents and query, a technique known as *cross-encoder*.

**End-to-end retrieval.**    Most of the methods described in the previous paragraph were used to re-rank a small number of documents, usually returned by a traditional IR systems. In the context of *ad-hoc* document retrieval, Gillick et al. (2018) showed that bi-encoder models could be competitive with traditional IR systems. For open domain question-answering, Karpukhin et al. (2020) introduced *dense passage retrieval* (DPR), which uses dense embeddings and nearest neighbors search. More precisely, question and passage embeddings are obtained using a BERT-based bi-encoder model, which is trained on a small dataset of question and passage pairs. Then, the full knowledge source (Wikipedia) is encoded using this model, and passages are queried by computing the k-nearest neighbors of the embedding of the question. Jointly embedding the query and documents makes the application of cross-encoder models intractable to large database. To address this limitation, Humeau et al. (2019) introduced the *poly-encoder* architecture, in which each documents is represented by multiple vectors instead of one. Similarly, Khattab et al. (2020) proposed a scoring function where each term of the query and documents is represented by a single vector. To make the method tractable, their system retrieves documents with an approximate score, which are then re-ranked with the exact one. Finally, Luan et al. (2020) conducts a theoretical and empirical study of sparse, dense and cross-attention information retrieval systems.

**Unsupervised learning.**    Closest to our work, there is growing body of work trying to learn information retrieval systems from unsupervised data. Lee et al. (2019) introduced the *inverse cloze task*

for pre-training retrievers, which can then be fine-tuned end-to-end on question-answering tasks. This pre-training scheme was later evaluated for ad-hoc document retrieval by Chang et al. (2020). Guu et al. (2020) proposed to augment language model pre-training with a retriever module, which is trained using the masked language modeling objective. Similarly, Lewis et al. (2020a) introduced a sequence-to-sequence model that is pre-trained by generating a target text, after retrieving a set of related texts. Lewis et al. (2020b) further train the retriever obtained in Karpukhin et al. (2020) by backpropagating to the retriever the error between the generated output and the gold answer.

Simultaneously to our work, Yang & Seo (2020) proposes to train a retriever with knowledge distillation. The main difference with our method is the nature of the synthetic labels that are used to train the retriever. Yang & Seo (2020) uses the DPR reader, which includes a classifier that predicts which passage contains the answer, and can be seen as a cross-encoder reranker. This technique thus performs the distillation of a cross-encoder retriever to a bi-encoder retriever. In contrast, our method uses the internal attention scores of the reader, which does not require additional supervision besides pairs of question and answer.

## 3 METHODOLOGY

Our system is composed of two modules, the retriever and the reader, following the standard pipeline for open-domain question answering. Given an input question these modules are used in a two-step process to generate an answer. First the retriever selects support passages in a large knowledge source. Then these passages are processed by the reader, along with the question, to generate an answer. For the reader module we use the Fusion-in-Decoder model (Izacard & Grave, 2020), which achieves state-of-the-art performance when combined with BM25 or DPR (Karpukhin et al., 2020). It is based on a sequence-to-sequence architecture, and is initialized from pre-trained models such as T5 or BART (Raffel et al., 2019; Lewis et al., 2019).

The focus of this work is to train the retriever without strong supervision or weakly supervised learning based on heuristics. For this we propose to train the retriever by learning to approximate the attention score of the reader. The training scheme outlined here can be seen as a student-teacher pipeline, where the teacher, the reader module, produces targets which are used to train a student network, the reader. By doing so, we hope to leverage the signal extracted from the question-answer pairs by the reader. Since the goal of the retriever is to retrieve the most relevant passages, by training the retriever to estimate the reader attention scores, we implicitly make the assumption that these scores are a good proxy for the usefulness of a passage to answer the question.

In this section we will first describe the Fusion-in-Decoder architecture, before elaborating on the signal which is used to train the retriever, the design of the retriever, and how this module is trained.

### 3.1 CROSS-ATTENTION MECHANISM

First, let us briefly review the Fusion-in-Decoder model (FiD, Izacard & Grave, 2020). The underlying architecture is a sequence-to-sequence model, composed of an encoder and a decoder. The encoder independently processes $n_p$ different text inputs $(s_k)_{1 \leq k \leq n_p}$. In the case of open-domain question answering based on Wikipedia, each input $s_k$ is the concatenation of the question $q$ and a support passage, with special tokens `question:`, `title:` and `context:` added before the question, the title of the Wikipedia article and the text of each passage. The output representations of the encoder are then concatenated to form a global representation $\mathbf{X}$ of dimension $(\sum_k \ell_k) \times d$, where $\ell_k$ is the length of the $k$-th segment and $d$ is the dimension of the embeddings and hidden representations of the model. Then, the decoder processes this representation as a regular autoregressive model, alternating self-attention, cross-attention and feed-forward modules.

Only the cross-attention module explicitly takes as input the global output representation $\mathbf{X}$ of the encoder. If $\mathbf{H} \in \mathbb{R}^d$ denotes the output of the previous self-attention layer of the decoder, the cross-attention operation consists in the following operations. First, queries $\mathbf{Q}$, keys $\mathbf{K}$ and values $\mathbf{V}$ are computed by applying linear transformations:

$$\mathbf{Q} = \mathbf{W}_Q \mathbf{H}, \quad \mathbf{K} = \mathbf{W}_K \mathbf{X}, \quad \mathbf{V} = \mathbf{W}_V \mathbf{X}.$$

Then a similarity score between the query at position $i$, $\mathbf{Q}_i$, and the key at position $j$, $\mathbf{K}_j$, is obtained by computing the dot-product between these two elements, and normalized over the dimension:

$$\alpha_{i,j} = \mathbf{Q}_i^T \mathbf{K}_j, \qquad \tilde{\alpha}_{i,j} = \frac{\exp(\alpha_{i,j})}{\sum_m \exp(\alpha_{i,m})}.$$

A new representation is obtained as a sum of the values, weighted by the attention probabilities, before going through a final linear transformation $\mathbf{W}_o$:

$$\mathbf{O}_i = \mathbf{W}_O \sum_j \tilde{\alpha}_{i,j} \mathbf{V}_{i,j}$$

The operations described above are performed in parallel with different linear transformations in the case of multi-head attention. Finally a normalization layer is applied, and this pipeline is wrapped by a skip connection. See Vaswani et al. (2017) for more details on the structure of Transformers.

### 3.2 CROSS-ATTENTION SCORE AS A RELEVANCE MEASURE FOR PASSAGE RETRIEVAL

In some sense, the attention scores $\alpha_{:,j}$ involving the $j$-th key measures the importance of this key, and corresponding value, to compute the next representation. We hypothesize that it is good proxy to estimate the relevance of a passage — the more the tokens in a text segment are attended to, the more relevant the text segment is to answer the question.

Given the reader model, an input question $q$ and a corresponding set of support passages $\mathcal{D}_q = (p_k)_{1 \le k \le n}$, we obtain relevance scores $(G_{q,p_k})_{1 \le k \le n}$ for each passage by aggregating attention scores. In particular, the score $G_{q,p_k}$ is obtained by averaging the pre-attention scores $\alpha_{0,:}$ over all the tokens in the input $s_k$ corresponding to the passage $p_k$, all the layers and all the heads of the decoder. Note that the FiD decoder jointly processes the passages, and thus the score $G_{q,p_k}$ depends on the other support passages. We consider other pooling operators, such as *max*, to aggregate attention scores over layers, heads and tokens and empirically compare them in Sec. 5.2.

Before we proceed, let us consider the following simple experiment, which is a first indication that reader attention scores are indeed a strong relevance signal. Given a question and 100 passages retrieved with DPR, our goal is to select the 10 best passages. When using the top 10 passages from DPR instead of the top 100, the performance of our reader drops from 48.2 EM to 42.9 EM. On the other hand, if we select the top 10 documents according to the attention scores, the performance only drops to 46.8 EM.

### 3.3 DENSE BI-ENCODER FOR PASSAGE RETRIEVAL

Ideally, we would like to rank passages according to the reader cross-attention scores. In practice however, since the passages and the question need to be processed simultaneously by the reader module it is impractical to query a large knowledge source this way. Thus, we use a retriever model composed of an embedder function $E$ that maps any text passage to a $d$-dimensional vector, such that the similarity score between a question $q$ and a passage $p$ is defined as $S_\theta(q, p) = E(q)^T E(p)$. This similarity metric enables us to index all passages in the knowledge source as a preprocessing step. Then at runtime, passages with the highest similarity score with the input question are retrieved, by using an efficient similarity search library such as FAISS (Johnson et al., 2019).

For the embedder we use BERT and follow DPR by considering that the encodings $E(q)$ and $E(p)$ are obtained by extracting the representation of the initial `[CLS]` token. This leads to a representation of dimension $d = 768$ in the case of a base model. Differently from DPR, we use the same encoding function $E$ for the questions and passages by sharing parameters.

### 3.4 DISTILLING THE CROSS-ATTENTION SCORE TO A BI-ENCODER

In this section, we describe how to train the retriever model, based on the relevance scores obtained in Sec. 3.2. For the training objective of the retriever, we propose to minimize the KL-divergence between the output $S_\theta(q, p)$ and the score $G_{q,p}$ after normalization:

$$\mathcal{L}_{\text{KL}}(\theta, \mathcal{Q}) = \sum_{q \in \mathcal{Q}, p \in \mathcal{D}_q} \tilde{G}_{q,p}(\log \tilde{G}_{q,p} - \log \tilde{S}_\theta(q, p)),$$

where

$$\tilde{G}_{q,p} = \frac{\exp(G_{q,p})}{\sum_{p' \in \mathcal{D}_q} \exp(G_{q,p'})}, \qquad \tilde{S}_\theta(q,p) = \frac{\exp(S_\theta(q,p))}{\sum_{p' \in \mathcal{D}_q} \exp(S_\theta(q,p'))}.$$

In Sec. 5.1 we present results obtained when using alternatives to this training objective. We consider two other objectives which have been used in Dehghani et al. (2017), where BM25 is used as a teacher model to train a neural ranker. A first option consists in training the retriever with a regression approach by minimizing the mean squared error:

$$\mathcal{L}_{\text{MSE}}(\theta, \mathcal{Q}) = \sum_{q \in \mathcal{Q}, p \in \mathcal{D}_q} (S_\theta(q,p) - G_{q,p})^2.$$

The second option we consider is to use a max-margin loss that explicitly penalizes inversions in the ranking estimated by the retriever:

$$\mathcal{L}_{\text{ranking}}(\theta, \mathcal{Q}) = \sum_{q \in \mathcal{Q}, p_1, p_2 \in \mathcal{D}_q} \max\left(0, \gamma - \text{sign}(G_{q,p_1} - G_{q,p_2})(S_\theta(q,p_1) - S_\theta(q,p_2))\right).$$

In words, if $p_1$ is more relevant to answer the question $q$ than $p_2$, i.e. $G_{q,p_1} > G_{q,p_2}$, the loss pushes the retriever score of $p_1$ to be larger than the score of $p_2$ by at least a margin of $\gamma$.

### 3.5 ITERATIVE TRAINING

In this section, we explain how iterative training can be used with the student-teacher scheme described in the previous section, similarly to Khattab et al. (2020). This iterative procedure can be interpreted as using the current retriever to sample negative examples, in order to train a new retriever. When learning a retriever with discriminative training, negative samples play an important role, and various strategies have been considered in previous work. Karpukhin et al. (2020) compared random sampling with using the top-k passages from BM25 which do not contain the answer and with using the positive passages from other queries. Consider that for each question, we have an initial set of support documents $\mathcal{D}_q^0$. We propose to use an iterative pipeline where each iteration can be described as the following 4-step process:

1. Train the reader $R$ using the set of support documents for each question $\mathcal{D}_q^0$.
2. Compute aggregated attention scores $(G_{q,p})_{q \in \mathcal{Q}, p \in \mathcal{D}_q^0}$ with the reader $R$.
3. Train the retriever $E$ using the scores $(G_{q,p})_{q \in \mathcal{Q}, p \in \mathcal{D}_q^0}$.
4. Retrieve top-passages with the new trained retriever $E$.

This multi-step procedure can be repeated multiple times. A critical point of the training procedure is the initial set of documents corresponding to each question. In Sec. 4, we compare retrievers obtained by starting from documents obtained using BM25 or cosine similarity from a BERT model. In particular, we show that while the initial performance with BERT is low, the iterative procedure allows to greatly improve the performance of the model.

## 4 EXPERIMENTS

In this section we evaluate the student-teacher training procedure from the previous section. We show that we obtain competitive performance without strong supervision for support documents.

### 4.1 EXPERIMENTAL SETTING

**Datasets.** We perform experiments on TriviaQA (Joshi et al., 2017) and NaturalQuestions (Kwiatkowski et al., 2019), two standard benchmarks for open-domain question answering. TriviaQA is made of questions from trivia and quiz league websites, and does not contain gold support documents. NaturalQuestions contains questions corresponding to web search queries, and gold support documents from Wikipedia. Following the setting from Lee et al. (2019); Karpukhin

| | Iter. | BERT | | | BM25 | | |
| --- | --- | --- | --- | --- | --- | --- | --- |
| | | P@20 | P@100 | Dev EM | P@20 | P@100 | Dev EM |
| NaturalQuestions | 0 | 4.8 | 12.0 | 9.8 | 59.3 | 74.0 | 41.2 |
| | 1 | 32.2 | 45.8 | 16.9 | 76.4 | 84.3 | 46.8 |
| | 2 | 51.1 | 62.6 | 28.6 | 80.4 | 86.7 | 47.9 |
| | 3 | 67.8 | 76.8 | 39.3 | 80.0 | 86.3 | 46.2 |
| TriviaQA | 0 | 4.6 | 12.0 | 9.7 | 75.0 | 82.3 | 65.3 |
| | 1 | 37.1 | 59.4 | 19.6 | 79.0 | 85.5 | 66.7 |
| | 2 | 60.8 | 73.4 | 43.3 | 82.1 | 86.5 | 67.5 |
| | 3 | 72.0 | 83.2 | 52.0 | 81.6 | 86.6 | 67.7 |
| | 4 | 76.4 | 84.6 | 62.3 | - | - | - |

Table 1: Iterative training starting with documents retrieved with BERT and BM25. Iteration 0 corresponds to the performance of the reader trained on the set of initial support documents. We report all metrics on the validation set.

et al. (2020), we use the original evaluation set as test set, and keep 10% of the training data for validation. We use the Wikipedia dump from Dec. 20, 2018 for support documents, splitting articles into non-overlapping passages of 100 tokens, and applying the same preprocessing as Chen et al. (2017).

We also evaluate on NarrativeQuestions (Kočiský et al., 2018), using a publicly available preprocessed version.[1] This is a reading comprehension dataset built on a corpus of books and movie scripts. For each story, questions are generated by human annotators based on a summary of the given document. We consider the full story setting, where the task is to answer questions given the entire story and not the summary used to generate question-answer pairs. Here the knowledge source is not the same for all questions: given a question the retrieval operation is performed on all passages of the associated story. These passages are obtained by dividing the story in chunks of 100 words. These stories are long documents, with an average of 60k words. While part of the documents could be processed entirely by the Fusion-in-Decoder module, it is interesting to limit the number of support passages to reduce the computational cost of the reading step.

While answers in TriviaQA and NaturalQuestions are short, NarrativeQA answers are about five words long on average, with medium length answers such as *"He dismantles it and attaches it to his mother's jeep"* which answers the question *"What does Mark do with his radio station?"*. Notably a significant number of answers do not correspond to spans in the story. It is thus not straightforward to train the retriever with heuristics using question-answer pairs. In our case we use the same pipeline as for TriviaQA and NaturalQuestions, demonstrating the flexibility of our approach.

**Evaluation.** The model performance is assessed in two ways. First, following previous work such as DPR and ColbertQA, we report the top-$k$ retrieval accuracy (P@k), which is the percentage of questions for which at least one passage of the top-$k$ retrieved passages contains the gold answer. It is unclear how well this metric evaluates the retriever performance, since the answer can be contained in a passage without being related to the question. This is notably true for common words or entities.

We also report the final end-to-end performance of the question answering system composed of the retriever and reader modules. This is the metric we are fundamentally interested in. For TriviaQA and NaturalQuestions, predicted answers are evaluated with the standard exact match metric (EM), as introduced by Rajpurkar et al. (2016). For NarrativeQA we report the metrics proposed in the original paper: ROUGE-L, BLEU-1, BLEU-4 and METEOR.

## 4.2 TECHNICAL DETAILS

**Initialization.** Similarly to DPR, we initialize the retriever with the BERT base model, pretrained with uncased text. The Fusion-in-Decoder reader is initialized with the T5 base model. A critical component of the iterative training procedure is the initialization of the support passages $\mathcal{D}_q^0$ asso-

---

[1] https://cs.nyu.edu/~kcho/NarrativeQA

| Model | NQ | | TriviaQA | |
|---|---|---|---|---|
| | dev. | test | dev. | test |
| DPR (Karpukhin et al., 2020) | - | 41.5 | - | 57.9 |
| RAG (Lewis et al., 2020b) | - | 44.5 | - | 56.1 |
| ColBERT-QA (Khattab et al., 2020) | - | 48.2 | - | 63.2 |
| Fusion-in-Decoder (T5 base) (Izacard & Grave, 2020) | - | 48.2 | - | 65.0 |
| Fusion-in-Decoder (T5 large) (Izacard & Grave, 2020) | - | 51.4 | - | 67.6 |
| Ours (starting from BERT, T5 base) | 39.3 | 40.0 | 62.5 | 62.7 |
| Ours (starting from BM25, T5 base) | 47.9 | 48.9 | 67.7 | 67.7 |
| Ours (starting from DPR, T5 base) | 48.0 | 49.6 | 68.6 | 68.8 |
| Ours (starting from DPR, T5 large) | **51.9** | **53.7** | **71.9** | **72.1** |

Table 2: Comparison to state-of-the-art models on NaturalQuestions and TriviaQA.

ciated with each question $q$. For this we consider different options. The first one is to use passages retrieved using BM25. We use the implementation from Apache Lucene[2] with default parameters, and tokenize questions and passages with SpaCy[3]. We also use passages obtained with BERT as a retriever without fine-tuning, this leads to poor initial performance. Finally in Table 2 we show that initializing $\mathcal{D}_q^0$ with passages obtained with DPR (Karpukhin et al., 2020) outperforms the two previous initializations. We train all retrievers using 100 passages. For the reader, we use 100 passages for NaturalQuestions and TriviaQA and 20 passages for NarrativeQA.

**Iterative training.** We apply the iterative training procedure on each dataset independently. Both the reader and the retriever are fine-tuned using the ADAM algorithm (Kingma & Ba, 2014), with a batch of size 64. The reader is trained for 10k gradient steps with a constant learning rate of $10^{-4}$, and the best model is selected based on the validation performance. The retriever is trained with a constant learning rate of $5 \cdot 10^{-5}$ until the performance saturates. To monitor the performance of the retriever during training, we measure the similarity between the reader and the retriever rankings. At each new training iteration the reader is reinitialized from T5 base, while we pursue the training of the retriever. We found that restarting from T5 base is important for the first iterations when starting with BERT documents. We have not tried to reinitialize the retriever between each iteration. More details on the hyperparameters and the training procedure are reported in Appendix A.2.

## 4.3 RESULTS

In Table 1, we report the performance of our approach for different number of self-training iterations. Generally, we observe that the accuracy of our system increases with the number of iterations, obtaining strong performance after a few iterations. Interestingly, while the initial performance with documents retrieved with BERT is very poor, our method still reach competitive scores on TriviaQA, and to a lesser extent, NaturalQuestions. However, a second observation is that the quality of the initial document sets plays an important role on the performance of the end system. Indeed, we observe that starting the procedure from BM25 documents, which are higher quality as indicated by the performance of the system at iteration 0, leads to stronger results than using BERT documents. An interesting research question would be to explore pre-training of the initial BERT model for retrieval, for example by using the inverse cloze task.

In Table 2, we report the performance of our approach, as well as existing state-of-the-art systems on TriviaQA and NaturalQuestions. In addition to initializing our method with documents retrieved with BM25 and BERT, we also train a system by starting from DPR documents. First, we observe that our method improve the performance over the state-of-the-art, even when starting from BM25 documents. This validates our assumption that it is possible to obtain strong retrievers without the need of supervision for the documents. Second, when starting from DPR passages, our method leads to a +4.5 EM improvement on TriviaQA and +2.3 EM improvement on NaturalQuestions when the final evaluation is carried out with a large reader.

---

[2]`lucene.apache.org`
[3]`spacy.io`

| Method | Iter. | Rouge-L | | Bleu-1 | | Bleu-4 | | Meteor | |
|---|---|---|---|---|---|---|---|---|---|
| | | dev. | test | dev. | test | dev. | test | dev. | test |
| Best from Kočiskỳ et al. (2018) | - | 14.5 | 14.0 | 20.0 | 19.1 | 2.23 | 2.1 | 4.6 | 4.4 |
| DPR + FiD | - | 29.7 | 30.8 | 33.0 | 34.0 | 6.7 | 6.9 | 10.3 | 10.8 |
| Ours starting from BM25 | 0 | 29.9 | 30.3 | 34.6 | 33.7 | 7.1 | 6.5 | 10.5 | 10.4 |
| Ours starting from BM25 | 1 | **31.6** | **32.0** | **34.9** | **35.3** | **7.6** | **7.5** | **11.0** | **11.1** |

Table 3: Performance on NarrativeQA.

In Table 3, we report the performance of our method on the NarrativeQA dataset. We use the setting where the knowledge source corresponds to the whole document, and in particular, we do not use the summary. We compare our results to the best ones reported in the original paper for this setting. Similar to results obtained on NaturalQuestions and TriviaQA, we observe that training the retriever by using the attention scores of the reader leads to improvements, compared to the BM25 baseline.

## 5 ABLATIONS

In this section, we investigate design choices regarding two key elements of our approach: the training objective and the aggregation of cross-attention scores. For all experiments, we consider a simplified experimental setting: a single training iteration is performed on NaturalQuestions, starting from BM25 passages.

### 5.1 TRAINING OBJECTIVES

In Table 4 we report the performance of our model trained with the different training objectives described in Sec. 3.3. We observe that using the KL-divergence between the aggregated scores of the reader and the scores of the retriever outperforms the other objective functions.

| Method | P@5 | P@20 | P@100 | Dev EM |
|---|---|---|---|---|
| Mean Squared Error | 46.5 | 61.2 | 73.9 | 40.6 |
| Max-margin loss, $\gamma = 1$ | 60.3 | 73.6 | 82.7 | 45.4 |
| Max-margin loss, $\gamma = 0.2$ | 60.3 | 73.5 | 82.6 | 45.8 |
| Max-margin loss, $\gamma = 0.1$ | 60.2 | 73.5 | 82.6 | 45.1 |
| KL-divergence | 64.7 | 76.4 | 84.3 | 46.8 |

Table 4: Comparison of training objectives on NaturalQuestions after one iteration. We report all the metrics on the validation set.

### 5.2 HOW TO AGGREGATE CROSS-ATTENTION SCORES?

In Section 4 the cross-attention scores $\alpha$ are aggregated in a specific way, in order to obtain a single scalar used to train the retriever. Formally let us denote by $\alpha_{i,j,k,h}$ the cross-attention scores between token $i$ of the output and token $j$ of the input, for the $k$-th layer and $h$-th head. Then, the scores $G_{q,p}$ for $p \in \mathcal{D}_q$ used in Section 4 are computed as follows:

$$G_{q,p} = \underset{j,k,h}{\text{mean}}\, \alpha_{0,j,k,h},$$

where $j$ describes the input tokens corresponding to $p$. In Table 5 we explore alternatives to this choice by considering different aggregation schemes. In particular, we consider (1) taking the max over the input tokens corresponding to passage $p$ instead of the average, (2) taking the average over the output tokens instead of taking the score of the first token, (3) taking the mean over the last six layers instead of all the layers, (4) taking the max over the layers instead of the average, (5) taking the max over the heads instead of the average. We observe that the performance of our approach is relatively stable to the choice of aggregation, and that the best result is obtained by averaging, except over the output tokens where it is best to only consider the first token.

| Method | P@5 | P@20 | P@100 | Dev EM |
|---|---|---|---|---|
| (0) $\mathrm{mean}_{j,k,h}\,\alpha_{0,j,k,h}$ | 64.7 | 76.4 | 84.3 | 46.8 |
| (1) $\mathrm{mean}_{k,h}\,\max_j\,\alpha_{0,j,k,h}$ | 61.2 | 72.5 | 81.0 | 46.0 |
| (2) $\mathrm{mean}_{i,j,k,h}\,\alpha_{i,j,k,h}$ | 63.5 | 75.3 | 83.1 | 45.8 |
| (3) $\mathrm{mean}_{7\leq k\leq 12,j,h}\,\alpha_{0,j,k,h}$ | 64.1 | 75.7 | 83.8 | 46.4 |
| (4) $\mathrm{mean}_{j,h}\,\max_k\,\alpha_{0,j,k,h}$ | 63.9 | 75.5 | 83.7 | 46.5 |
| (5) $\mathrm{mean}_{j,k}\,\max_h\,\alpha_{0,j,k,h}$ | 64.2 | 76.1 | 83.9 | 46.8 |

Table 5: Comparison of attention aggregation schemes on NaturalQuestions after one iteration. The index $i$ corresponds to output tokens, $j$ corresponds to input tokens of a given passage, $h$ to heads and $k$ to layers of the decoder. We report all metrics on the validation set.

## 6 CONCLUSION

In this paper, we introduce a method to train an information retrieval module for downstream tasks, without using pairs of queries and documents as annotations. Our approach is inspired by knowledge distillation, where the retriever module corresponds to the student model and the reader module corresponds to the teacher model. In particular, we use the cross-attention scores, from a sequence-to-sequence reader, to obtain synthetic targets for the retriever. We compare different ways to aggregate the scores, as well as different training objectives to learn the retriever. We show that iteratively training the reader and the retriever leads to better performance, and obtain state-of-the-art performance on competitive question answering benchmarks. In the future, we would like to explore better pre-training strategies for the retriever module, as well as better scoring functions for the retriever.

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

| Hyperparameter | Reader-base | Reader-large | Retriever |
|---|---|---|---|
| Number of parameters | 220M | 770M | 110M |
| Number of heads | 12 | 16 | 12 |
| Number of layers | 24 | 48 | 12 |
| Hidden size | 768 | 1024 | 768 |
| Batch size | 64 | 64 | 64 |
| Dropout | 0.1 | 0.1 | 0.1 |
| Learning rate schedule | constant | linear | constant |
| Peak learning rate | 0.0001 | 0.00005 | 0.00005 |
| Gradient clipping | 1. | 1. | 1. |

Table 6: Hyperparameters for retriever and reader training.

# A  EXPERIMENTAL DETAILS

## A.1  SETTING

For NaturalQuestions and TriviaQA we follow the standard open-domain question answering setting used in Lee et al. (2019); Karpukhin et al. (2020). In this setting the original development set is used as test set, and 10% of the training set is used for development purpose. Moreover, for NaturalQuestions, all questions with answers longer than five tokens are discarded.

For TriviaQA we use the unique human-generated answer to train the reader. In this dataset part of the answers are in uppercase. We normalize uppercase answers by converting the first letter in each word to uppercase and remaining characters to lowercase using the `title` Python string method.

For NarrativeQA, questions and answers in uppercase are converted to lowercase.

## A.2  TRAINING

For every datasets, both the reader and the retriever are fine-tuned with a dropout rate of 10%. All models at the exception of the large reader are trained using the ADAM algorithm (Kingma & Ba, 2014) with a constant learning rate of $10^{-4}$ for the base reader and $5 \cdot 10^{-5}$ for the retriever. The base reader is trained for 10k gradient steps with a batch size of 64. We train the large reader with the ADAMW algorithm (Loshchilov & Hutter, 2019) with a peak learning rate of $5 \cdot 10^{-5}$ and a linear warmup for 600 gradient steps followed by a linear decrease of the learning rate for 14.4k gradient steps.

We perform model selection on the validation performance. The retriever is trained until its performance saturates with a batch size of 64. To monitor the performance of the retriever during training, we measure the similarity between the ranking obtained with the reader score, and the ranking of the retriever. We use different metrics for this: the number of inversions between the two rankings, the proportion of passages in the retriever top-$k$ that are also in the reader top-$k$ and the number of passages to obtain all top-$k$ passage of the reader.

During training and at test time, each text input of the encoder is restricted to be at most 250 token long. For NaturalQuestions and TriviaQA, we use wikipedia as a knowledge source, thus for each passage there is an associated article title. Each input is composed of the concatenation of a question, title and support passage with special tokens `question:`, `title:` and `context:` added before the question, the title and the text of each passage. In the case of NarrativeQA, the question and each passage are concatenated to form the different inputs.

## A.3  INFERENCE

At test time, for TriviaQA and NaturalQuestions we use greedy decoding, and Beam Search with 3 beams for NarrativeQA.

| Iter. | NaturalQuestions | | | TriviaQA | | |
|---|---|---|---|---|---|---|
| | P@20 | P@100 | Dev EM | P@20 | P@100 | Dev EM |
| 0 | 77.1 | 84.3 | 46.4 | 78.2 | 84.7 | 65.0 |
| 1 | 80.3 | 86.7 | 47.8 | 81.4 | 86.4 | 67.1 |
| 2 | 82.4 | 87.9 | 48.2 | 83.5 | 87.4 | 68.1 |

Table 7: Iterative training starting with documents retrieved with DPR. Iteration 0 corresponds to the performance of the reader trained on the set of initial support documents. We report all metrics on the validation set. Contrary to results reported in Table 1, the reader model was not re-initialized between each iteration.

