# OpenReview forum: "Distilling Knowledge from Reader to Retriever for Question Answering"
_ICLR.cc/2021/Conference — ICLR 2021 Poster_

### Official Review · AnonReviewer2 · 2020-10-25
**An interesting contribution to supervised training of retriever models**

**Rating:** 7
**Confidence:** 3

**Review:**

Many Information Retrieval systems rely on two components: a retriever that identifies a small set of "support" documents from a large corpus, followed by a reader that re-scores these support documents more finely. For retrievers, metrics like BM25 were once common but they are increasingly replaced by machine learned components. However, most datasets do not provide direct supervision information for the retriever.

Assuming a Transformer reader, this paper proposes to train the reader and retriever iteratively, using cross-attention scores from an increasingly better reader as a way to identify increasingly better sets of support documents. The novelty of this paper is the use of reader cross-attention scores as a proxy to train the retriever. The models are otherwise standard.

This paper is well motivated and clearly written. It seems to be improving the state-of-the-art on three datasets and over several recent baselines. The training procedure (including hyper-parameters) is detailed, facilitating reproducibility.

Questions and comments:
- Since DPR plays a central role in this paper, it deserves a longer introduction, either in Section 2 or later. Further discussion on the use of already trained models as starting point would be interesting.
- Similarly, it would be nice to summarize the negative example sampling strategy in Section 3.5.
- On the NarrativeQA dataset, the authors mention "it is not straightforward to train the retriever with heuristics using Q-A pairs". Isn't BM25 (used for the initial set of documents) a heuristic? Can this be clarified?
- I assume the gains against the baselines are statistically significant, but it would be nice to mention in writing.
- An iterative training procedure is going to be slower than other approaches. Some discussion on this point would be interesting. Or referencing a discussion in a previous paper would also be OK.
- Are there any other downsides to this approach?

Minor comments:
- In the third paragraph of the first section, it sounds like the authors introduce the retriever/reader architecture, which is standard (and otherwise clear from the rest of the paper).
- Typo, page 2: well know -> well known
- Please introduce d (embedding dimensionality) in page 3.
- Please add a citation in the first row of Table 3.

---

> ### Author Response · Authors · 2020-11-19
> **Reply to Reviewer 2**
>
> We thank R2 for the useful comments.
>
> We added an introduction of the DPR method (Sec. 2), as well as the negative sampling strategies (Sec. 3.5), and addressed minor comments in the revised version of the paper.
>
> **Isn't BM25 a heuristic?**
>
> What we meant by “heuristics” in that context is the way to obtain positive passages from QA pairs. For datasets without hand annotated passages, but only pairs of question and answer, previous works considered a passage to be a positive example for training the retriever if it contains the answer. We argue that for long form question answering, such as NarrativeQA or ELI5, such heuristics cannot be used, as many answers do not appear in passages.
>
> **Slower than other approaches**
>
> When starting from (reasonably) good passages, such as DPR or BM25, iterative training is not significantly slower than other approaches. Indeed, one does not need to train the reader from scratch at each iteration (similarly, we do not re-initialize the retriever between iterations). Thus each iteration of training is less expensive than training a retriever and a reader from scratch. In practice, when starting with DPR or BM25 passages we overall train the retriever for roughly the same number of epochs as DPR (30). We believe a potential avenue to improve the training procedure would be to have a joint training of the two models, instead of an iterative procedure.

---

### Official Review · AnonReviewer3 · 2020-10-26
**Official Blind Review #3**

**Rating:** 7
**Confidence:** 4

**Review:**

Paper Summary:
* This paper proposes a technique to learn retriever models for question answering that does not require annotated pairs of query and documents. The proposed technique uses  attention scores of a reader model to obtain synthetic labels for the retriever.  Experimental results with NaturalQuestions, TriviaQA, and NarrativeQA show that the technique achieved state-of-the-art results.

Strengths:
* This paper is well organized and well written.
* The proposed method is novel.
* Experimental results are strong.

Comments:
* We can see from Table 1 of this paper and Table 2 of the DPR paper that the P@20 and P@100 scores of the proposed retriever, which used iterative training starting with documents retrieved with BM25, outperformed those of the DPR model trained in a supervised manner.  This is a surprising result. I think this paper can benefit a lot with a more analysis with this point.  How much false positives and negatives are included in the reader's attention scores? Why are the readers' attention scores better than supervised  labels?
* I also would like to see the results of the proposed system that uses supervised labels of relevant passages (e.g. a cross entropy loss) in addition to the KL divergence loss.  Moreover, I would like you to add the results of iterative training starting with documents retrieved with DPR to Table 1.
* There is a concurrent study.  I think it would be very useful to discuss the study: "Is Retriever Merely an Approximator of Reader?" https://openreview.net/forum?id=dvXFpV6boX

Update:
Thank you for the answers to my questions and additional experiments!

---

> ### Author Response · Authors · 2020-11-19
> **Reply to Reviewer 3**
>
> First, we would like to thank the reviewer for the comments and questions!
>
> **Comparison with DPR**
>
> On NaturalQuestions, which contains gold annotated passages used to train DPR, multiple reasons might explain why our method performs competitively. First, for each question, DPR uses only one positive example, while our approach might benefit from multiple passages containing the answer (we give examples of such passages below). Second, the iterative training procedure might lead to higher quality negative examples than the ones obtained from the strategies used by DPR (obtained using BM25, or using the positive passages of other questions). Please note that on TriviaQA, there are no gold annotated passages, and DPR relies on a heuristics, that the answer is contained in a passage, to obtain positive examples. This might also lead to false positive. Finally, in DPR, when no positive passages were found, the question was discarded, while we keep the full training set to train our retriever. Please also note that DPR report results on the test set, while we report results on the dev set in Table 1.
>
> **Training a retriever with a cross entropy loss**
>
> We trained additional retrievers following your suggestion to add a supervised classification loss for relevant passages. We used the cross entropy loss, between the prediction of our retriever and supervised labels. As supervision all passages of the top100 of BM25 containing the answer are tagged as positive, and the rest of the BM25-retrieved passages are used as negative. We combine this loss with the KL-loss between the predicted scores and the attention scores to train a new retriever. This deteriorates performance: on the dev set of NaturalQuestions: we obtain P@5: 61.4, P@20: 72.8, P@100: 81.7 versus P@5:64.7, P@20: 77.3, P@100: 84.5 without the additional loss component. We also tried to use a single positive label, either by selecting the highest ranked passage of BM25 that contains the answer, or by using the human annotated passage used to train the DPR model. This leads to a more significant performance deterioration. Intuitively this comes from the fact that this training data contains false negatives, as passages retrieved with DPR or BM25 might also contain the answer. Results with supervised loss might be improved by considering better negative sampling strategies, but this would mean that the two losses are trained on different sets of passages.
>
> **Comparison with the concurrent ICLR submission: "Is Retriever Merely an Approximator of Reader?"**
>
> While both papers use distillation to train the retriever, there is an important distinction between the two, which is the source of the synthetic labels. In the concurrent submission, the authors propose to distill a supervised cross-encoder retriever to a bi-encoder retriever (thus requiring supervision for the teacher model, in the form of labeled passages). On the other hand, our main contribution is to use the reader attention activations to train the retriever, which does not require passage labelling.
>
> **Modifications in the revised version**
>
> We have added the results of iterative training starting from documents retrieved with DPR in Table 7 of the appendix, as we did not use the same exact training procedure as for Table 1: the difference is that the reader model was not re-initialized between each iteration of the iterative training.
> A paragraph discussing the difference with the concurrent study "Is Retriever Merely an Approximator of Reader?" (https://openreview.net/forum?id=dvXFpV6boX) has also been added to the revised version of the paper.

---

> > ### Author Response · Authors · 2020-11-19
> > **Examples of questions with multiple relevant passages**
> >
> > **Examples of questions with multiple relevant passages**
> >
> > Question: when did the supreme court hand down the brown v. board of education ruling
> > Answer: *May 17 , 1954*
> > P1: Brown v. Board of Education of Topeka, 347 U.S. 483 (1954), was a landmark United States Supreme Court case in which the Court declared state laws establishing separate public schools for black and white students to be unconstitutional. The decision effectively overturned the "Plessy v. Ferguson" decision of 1896, which allowed state-sponsored segregation, insofar as it applied to public education. Handed down on **May 17, 1954**, the Warren Court's unanimous (9–0) decision [...]
> > P2: [...] The Supreme Court declared portions of "passive resistance" unconstitutional in 1964 and again in 1968. On **May 17, 1954**, the U.S. Supreme Court handed down its ruling in "Brown v. Board of Education", in which the unanimous court held that separate public schools for black and white students was unconstitutional. [...]
> >
> > Question: what is the parent company of tj maxx
> > Answer: *TJX Companies*
> > P1: TJ Maxx is an American department store chain, selling at prices generally lower than other major similar stores. It has more than 1,000 stores in the United States and Puerto Rico, making it one of the largest clothing retailers in the United States. The company is part of the **TJX Companies**, which also owns HomeGoods/HomeSense
> > P2: The **TJX Companies**, Inc. () is an American multinational off-price department store corporation, headquartered in Framingham, Massachusetts. It remained from the original Zayre Corp. that was established in 1956. Of its banners, HomeGoods, TJ Maxx, and Sierra Trading Post operate in the United States; [...]
> >
> > Question: which part of the earth is composed of the crust and upper mantle
> > Answer: *the lithosphere*
> > P1: [...] It is the top component of **lithosphere**: a division of Earth's layers that includes the crust and the upper part of the mantle. [...]
> > P2: [...] The crust and the cold, rigid, top of the upper mantle are collectively known as the lithosphere, and it is of the **lithosphere** that the tectonic plates are composed.

---

### Official Review · AnonReviewer4 · 2020-10-29
**The paper is about open domain QA. It describes an interesting approach to train the retriever using attention of the reader as weak supervision signal. This helps solve the problem of lack of QA-oriented text retrieval relevance judgments. The general assumption is that the reader can provide useful signals back to the reader. The experiments on 2 datasets confirm that such a process can indeed help increase the performance.**

**Rating:** 7
**Confidence:** 4

**Review:**

The paper targets an important problem in open-domain QA - the training of the retriever for the purpose of determining a segment that may contain the answer. In the most traditional setting, the retriever is just a traditional IR system such as BM25. In some existing work, the retriever has been trained to locate the documents containing the answer (e.g. inverse cloze task, or DPR). This paper goes in the same direction. The difference is that it uses the attention weights as relevance signals to train the retriever, instead of the inclusion of the answer in the passage.

Overall, the paper describes an interesting contribution to the problem. I vote for accepting it.

Strengths:
The proposed idea is interesting and complements those used in the existing work. In the experiments, it is shown that the idea can indeed improves the existing ones: it takes the existing work DPR as the starting point, and shows that the iterative process incorporating the attention weights can improve the QA performance.

The paper provides solid experiments that support the claim.

Weaknesses:
While the basic idea sounds interesting, the paper does not provide a strong intuition behind it. One could imagine that strong attention could be paid to wrong passages, as the reader does not know the ground truth. In contrast, the idea used in the previous work - the inclusion of the answer in a passage is a positive signal - sounds more intuitive. It would be interesting to estimate how the attentions correlate to with the distribution of correct answer in the passages. This could provide some empirical indication that attention weights point to the right direction. The paper contains Table 1, which partly show this correlation. A stronger intuitive motivation could make the paper stronger.

---

> ### Author Response · Authors · 2020-11-19
> **Reply to Reviewer 4**
>
> First, we would like to thank the reviewer for the comments and questions!
>
> **Inclusion of the answer in a passage is a positive signal**
>
> Regarding the idea used in previous work -considering that a passage containing the answer is a positive signal- we argue that this could also lead to false positives. Indeed, many answers correspond to frequent words or numbers (eg. years) that are contained in many passages. To illustrate our point, we’ve included examples of false positives (ie. passages containing the answer, but not being good evidence to answer the question), for which our reader attention scores are low (and thus ranked in the bottom 20 of the top 100 passages returned by BM25). Finally, please note that as the reader and retriever are trained on the same dataset, the reader had access to the answers when it was trained. Our method could thus be seen as a soft, or latent, version of the previously used heuristics.
>
> **Correlation between attention scores and the fact that a passage contains the answer**
>
> A way to measure that attention weights points to passages containing the answer is to rerank the top 100 of BM25 according to the attention score, and count the average number of passages containing the answer in the top-k passages after reranking. For the top-5, top-10, top-20, top-50 passages we obtain with attention scores: 1.47, 2.17, 3.12, 4.84 passages containing the answer, while for BM25 we have 0.80, 1.31, 2.13, 3.97. Similarly if we measure the ratio of questions for which reranking according to the attention scores the top 100 passages retrieved with BM25 gives at least a passage that contains the answer in the top-k documents, we have for the top-1, top-5, top-10: 44.3%, 63.2% and 70.6%, while BM25 gives 21.6%, 42.1% and 59.3%.
>
> **Example of false positives from BM25**
>
> Question: who won the most world cup in soccer
> Answer: *Brazil*
> False positive: [...] The CONMEBOL Beach Soccer Championship took place in Manta, Ecuador on 19–26 April 2015. A total of 10 teams took part in the tournament. The top three teams qualified for the 2015 FIFA Beach Soccer World Cup. **Brazil** and Paraguay won their respective semi-finals on 25 April 2015 to qualify for the World Cup. On the next day, Argentina won the third place match to also book a place in the World Cup, while **Brazil** defeated Paraguay in
>
> Question: who won the most world cup in cricket
> Answer: *Australia*
> False positive: [...] The 2009 Women's Cricket World Cup Final was a cricket match between New Zealand and England played on 22 March 2009 at the North Sydney Oval in **Australia**. It was the culmination of the 2009 Women's Cricket World Cup, the ninth Women's Cricket World Cup. England won the final by four wickets, clinching their third World Cup title and their first outside England. It was the second time that the two teams had met at this stage of a World Cup – England won their previous final contest in 1993. After winning the toss,
>
> Question: when was kingdom of the crystal skull filmed
> Answer: *2007*
> False positive: letter to his brother that he had recently acquired the skull in an auction from Burney, paying £400. Controversies continued when identical measurements were found between Sotheby's skull and Mitchell–Hedges' skull, leaving the authenticity of this artifact questionable at best. Mitchell-Hedges' crystal skull was retained in the possession of his adopted daughter until her death on 11 April **2007**. Prior to her death, the skull was only shown to the public periodically, making it hard for the skull to be accessed and tested for authenticity. However, since Anna's death the skull has been examined thoroughly and despite many previous claims

---

### Official Review · AnonReviewer1 · 2020-10-29
**Interesting technique**

**Rating:** 6
**Confidence:** 4

**Review:**

The authors propose a training technique for information retrieval models in the context of (open domain) question answering. Assuming the existence of some reader model, the idea is to use internal information of that model as a training signal for a retriever. Specifically, they use the attention activations over the input documents as synthetic labels for the retriever.

The paper is written well, and proposes an interesting idea. The technique is well motivated, I particularly like that they motivate the use of the cross-attention score through a simple experiment, where they compare the top 10 DPR passages with the top 10 passages by attention score.
The results over the SOTA seem moderate though, and the number of iterations seems to be an important (and potentially underexplored) variable there.

I support the acceptance of the paper, because I believe the technique and the choice of model score (cross-attention score) are both interesting contributions.

As the authors say, training retrieval systems is tricky, since there is usually not sufficient labelled data available and it might depend heavily on the task. The iterative training that exploits internal state of the “downstream” model that they describe is an interesting idea that deserves attention from the community.

In Table, only up to 4 iterations in the table, which still show large improvements from one to the next. It would be interesting to know at what point no additional gains are seen.

---

> ### Author Response · Authors · 2020-11-19
> **Reply to Reviewer 1**
>
> First, we would like to thank the reviewer for the comments and questions!
>
> While we acknowledge that the improvement is moderate on NaturalQuestions (+1.1 EM over SOTA), the improvement on TriviaQA is more significant (+3.9 EM over SOTA). This might be explained by the fact that manually annotated passages are provided with NQ and used to train DPR, while it’s not the case for TriviaQA, where the gold passage is selected with a heuristic. The TriviaQA setting corresponds to the main motivation of our paper, which is to train retriever systems without manually annotated passages.
>
> In table 4 we have stopped reporting numbers for the iterative process when there was no more improvement at the following iteration.

---

### Decision · Program_Chairs · 2021-01-07
**Final Decision**

**Decision:**

Accept (Poster)

**Comment:**

The paper attempts to improve retrieval in open domain question answering systems, which is a very important problem. In this regards, the authors propose to utilize cross-attention scores from a seq2seq reader models as signal for training retrieval systems. This approach overcomes typical low amount of labelled data available for retriever model. The reviewers reached a consensus that the proposed approach are interesting and novel. The proposed approach establish new state-of-the-art performance on three QA datasets, although the improvements over previous methods are marginal. Overall, reviewers agree that the paper will be beneficial to the community and thus I recommend an acceptance to ICLR.